# Inhibitory Synaptic Influences on Developmental Motor Disorders

**DOI:** 10.3390/ijms24086962

**Published:** 2023-04-09

**Authors:** Matthew J. Fogarty

**Affiliations:** Department of Physiology & Biomedical Engineering, Mayo Clinic, Rochester, MN 55902, USA; fogarty.matthew@mayo.edu

**Keywords:** motor neuron, GABA, glycine, spasticity, cerebral palsy, Rett syndrome

## Abstract

During development, GABA and glycine play major trophic and synaptic roles in the establishment of the neuromotor system. In this review, we summarise the formation, function and maturation of GABAergic and glycinergic synapses within neuromotor circuits during development. We take special care to discuss the differences in limb and respiratory neuromotor control. We then investigate the influences that GABAergic and glycinergic neurotransmission has on two major developmental neuromotor disorders: Rett syndrome and spastic cerebral palsy. We present these two syndromes in order to contrast the approaches to disease mechanism and therapy. While both conditions have motor dysfunctions at their core, one condition Rett syndrome, despite having myriad symptoms, has scientists focused on the breathing abnormalities and their alleviation—to great clinical advances. By contrast, cerebral palsy remains a scientific quagmire or poor definitions, no widely adopted model and a lack of therapeutic focus. We conclude that the sheer abundance of diversity of inhibitory neurotransmitter targets should provide hope for intractable conditions, particularly those that exhibit broad spectra of dysfunction—such as spastic cerebral palsy and Rett syndrome.

## 1. Introduction

The motor system is a complex assembly of the nervous system, skeleton and skeletal muscle in order to effect movement. Movement in the neuromotor sense includes life-sustaining behavioural activities such as ventilation [1] and mastication/deglutition [2] and more goal-oriented activities such as locomotion and execution of fine and gross motor skills [3,4,5,6]. In humans and many other animals, the neuromotor system is essential for communication [7].

The prime effector of the motor system is the motor unit, comprising an individual motor neuron (MN) and all the muscle fibres it innervates [8]. Motor units are generally classified as either slow or fast fatigue-resistant (type S and FR), or fast fatigueable (type FF), with the former comprising smaller MN innervating type I or IIa skeletal muscle fibres and the latter units comprising larger MNs innervating type IIx/b skeletal muscle fibres [9,10]. Recruitment of motor units is related to MN surface area, such that smaller MNs (with higher input resistance and lower capacitance) are recruited before larger MNs (low input resistance and higher capacitance) [11]. This phenomenon is known as Henneman’s Size Principle and underpins the gradations of force/torque production and fatigueability of muscular contractions to the desired behaviour [10].

Despite the cortical and pattern-generator (brainstem or spinal segment) influence on motor unit activation and the interactions of cerebellar and striatal regions on motor output, the MN is eminently capable of neural computation [12,13]. Indeed, the MN is the final common pathway, and receive excitatory (predominantly glutamatergic) inputs and both GABAergic and/or glycinergic inhibitory inputs, amongst a host of other extrinsic and intrinsic excitability modulators [14,15,16,17,18,19,20,21,22,23,24,25,26,27,28,29,30]. A simplified summary of neuromotor control shows key differences between respiratory and locomotor circuits (Figure 1), with respiratory circuits governed by a centralized pattern/rhythm generator (the pre-Bötzinger complex) [10,31] and locomotor circuits with distributed pattern generation at various segments of the brainstem and spinal cord [32,33,34,35].

It is well-established that a derangement of the balance between excitatory and inhibitory influences, specifically the impairment of synaptic inhibition can lead to a variety of conditions including epilepsy, autism spectrum disorders and schizophrenia [36]. In this review, we focus on the influence of GABAergic and/or glycinergic neurotransmission on motor disorders during development, including Rett syndrome [37] and spastic cerebral palsy, sCP [38]. We compare how Rett syndrome, with attention paid to both locomotor and respiratory motor symptoms and a widely adopted rodent model has an abundance of translatable information and therapeutic horizons, while sCP, with little focus on symptoms and no validated animal model remains plagued by limited understanding. We also outline how strategies aimed at restoring inhibitory activity or reducing excitatory/inhibitory imbalance may be useful in ameliorating symptoms or reducing disease morbidity.

## 2. Inhibitory Neurotransmission

### 2.1. GABA- and Glycinergic Neurotransmission

GABA (γ amino butyric acid) has a wide variety of synaptic (i.e., hyperpolarization) and extrasynaptic metabotropic effects on neurons [39,40,41,42,43,44,45], in addition to having influences on the immune system, the gastroenteric tract and in particular, pancreatic cells [46,47,48,49,50,51,52,53]. Ionotropic synaptic inhibition is predominantly mediated by transmembrane GABA-A receptors, with some GABA-C receptors present in certain brain regions. By contrast, metabotropic influences of GABA are mediated by GABA-B receptors [39,54]. In the present review, we will mainly focus on the postsynaptic effects mediated by ionotropic GABA-A receptors and drug interactions at GABA-B receptors (Figure 2).

GABA is synthesized from glutamate via the activity of the glutamate decarboxylase (GAD) whose two main isoforms are GAD65 and GAD67 in GABAergic interneurons [55,56,57]. In mature neocortex, hippocampul, cerebellar and spinal cord dorsal horn GABAergic neurons, GAD67 synthesizes the majority (~90%) of GABA within the cell, whilst GAD65 is found exclusively at presynaptic regions [58,59,60] (Figure 2). In terminal boutons within the cerebrum, striatum, cerebellum, and spinal cord, GABA is transported into presynaptic vesicles, by the vesicular inhibitory amino acid transporter (VGAT) prior to synaptic release [36,61,62,63]. In both the cortex and spinal cord, GABA reuptake from the synaptic cleft occurs at presynaptic terminals and astrocytes, with GABA transporter 1 (GAT1) predominating in the former, with astrocytic GAT2/3 providing ~20% of the total cycling [64,65,66,67] (Figure 2). GABA release influences the motor system, directly via synaptic inputs on MNs [16,68,69] and indirectly via synaptic inhibition of rhythm and motor pattern generators and supraspinal neuromotor circuits [70,71,72]. Direct GABAergic inhibition (neuronal hyperpolarisation) is particularly important in the more cranial regions of the central nervous system, such as the cortex, hippocampus, thalamus and brainstem [55,57,71], whereas presynaptic inhibition (i.e., the effect of GABA on the axon terminal, as opposed to the neuron) may play more of a role in the motor and sensory systems, particularly in the lumbar spinal cord [73,74,75,76,77]. Importantly, for the rest of this manuscript we will focus on direct GABAergic hyperpolarisation via GABA receptors, as opposed to presynaptic inhibition which diminishes the likelihood of excitatory presynaptic vesicle release.

Actions of GABA-A receptors are not uniform, with certain subtypes responsible for the synaptic fast acting phasic inhibition, with other subtypes mostly found in extrasynaptic tonically acting receptors [78,79]. The major subunit types of GABA-A receptors are the α, β, γ and δ, with each synaptic GABA-A receptor isoform a pentameric combination of subunits, with the majority containing two pairs of α/β dimers and one other subunit, usually γ [80] (Figure 2). Different GABA-A subtypes comprising different receptor subunit and splice variants exhibit marked pharmacological and pharmacokinetic variances. The major dimer α/β combination binds GABA [79,80], while the α/γ site binds GABA allosteric modulators such as benzodiazepines [81,82] (discussed later). Each subunit exhibits splice variants [80]; however, in homogenized mouse brain the majority of synaptic GABA-A receptors are α_1_β_2_γ_1_ [80,83], although α_2_ combinations more prevalent in the spinal cord [84] and MNs [85], with further complexity of isoforms likely to be evident during development. Extrasynaptic GABA-A receptors include a δ subunit, often with an α_4_ or α_6_, which have higher affinities to GABA compared to other α/β dimers [79] (Figure 2). In the hippocampus, extrasynaptic α_5_ subunits are involved in tonic inhibition and may also redistribute to synaptic areas in an activity-dependent manner [86]. Stabilisation of GABA-A receptors to the postsynaptic membrane is accomplished by Neuroligin-2 which form a trans-synaptic scaffold with presynaptic elements [87,88] (Figure 2).

### 2.2. Glycinergic Neurotransmission

Glycinergic inhibitory synaptic transmission is prevalent in the brainstem and spinal cord [17,21,22,89,90,91,92,93,94,95,96,97,98], although some other brain areas also demonstrate functional glycinergic transmission [90]. The ionotropic actions of glycinergic neurotransmission are mediated by the glycine receptor (Figure 2) and plays an important role in modulating motor patterns and motor circuit interneurons and MNs [90,99,100,101]. Despite presynaptic actions of glycine being an important modulator of neural activity, in this review, we will concern ourselves with synaptic inhibition of neurons via action at the glycine receptor, resulting in chloride mediated hyperpolarization [102] (Figure 2).

Glycine is synthesized by serine hydroxymethyltransferase within mitochondria, and like GABA, is bound in presynaptic vesicles by VGAT [36,61,62,63]. Subsequent to release, glycine reuptake from the synaptic cleft occurs at astrocytes and presynaptic terminals, with glycine transporter 1 (GlyT1) predominating in the former, with GlyT2 predominating at presynaptic release sites [103,104] and being the only reliable marker of glycinergic neurons [105] (Figure 2). In addition, the alanine–serine–cysteine-1 transporter (ASC1) also plays a role in astrocytic and neuronal reuptake [106,107], potentially distinct from those of GlyT1 and GlyT2 [105]. Corelease and or mixed release of GABA and glycine and inhibitory synapses is not uncommon within the brainstem and spinal cord [96,108,109]. As mentioned, the vesicular co-transporter of GABA and glycine, VGAT transports GABA and/or glycine to axon terminals prior to exocytosis—providing the mechanism for inhibitory co-release [36,61,62,63]. In cases where mixed release occurs, GABA predominates over glycine with regards to affinity to VGAT [61,62,110].

Glycine receptors are transmembrane proteins arranged as a pentamer of subunits (α [four], containing the ligand binding site and β) surrounding a pore, with the majority of fast inhibition occurring with the combinations of α_1_ and a β heteromer [102,111,112]. Alternative α subunits exhibiting altered pharmacokinetics [102,111], while the β subunit is responsible for interacting with gephyrin, which anchors the glycine receptor the synapse [113,114]. Beyond, glycine, glycine receptors do exhibit affinities for other amino acids and for GABA [115,116].

Clearly, from the complex interactions between GABA and glycine release, the nature of their receptors and their locations within the neuraxis there are many roles for inhibitory neurotransmission in the pathogenesis and/or treatment of various motor disorders. However, the elucidation of specific developmental disease or drug effects is hindered by the marked effects on neural and other organ development that GABA and glycine exert and the interplay between developmental chloride homeostasis and the depolarizing or hyperpolarizing actions of GABA and glycine at synapses.

### 2.3. Developmental Aspects of GABA and Glycine as Signalling Molecules and Neurotransmitters

GABA and glycine seem to have potent cell signalling effects during pre- and postnatal development independent of their role in synaptic neurotransmission. However, the precise nature of these cell-signalling pathways is obscured by the germline mutation loss of function approaches used in many developmental models of altered GABA and/or glycine. In some cases, elimination of foetal movement may be the trigger for morphological deformations [117,118]. Nonetheless, a variety of mutants examining the role of GABA and/or glycine synthesis, transport or receptor alterations have been used to parse out the distinct roles of altered signalling or altered synaptic activity on the development of the nervous system and other organs.

For GABA, GAD67 is the major GAD isoform in utero and GAD67 knockout mice exhibit marked ~90% loss of GABA [119], displaying cleft palate (~75%) and defects in the abdominal wall (omphalocele, ~45%) [119], while GAD65 knockout mice exhibit only modest ~20% GABA loss with no birth or growth abnormalities [120,121]. In agreement, rats with a deletion of the gene encoding GAD65 (*Gad2*) or the gene encoding GAD67 (*Gad1*), leads to GABA reduction and palate deformation [122], while double GAD65/GAD67 knockout mice exhibit an increased chance of cleft palate, omphalocele and abnormal spine curvature (kyphosis) [123]. In GAD67 knockout mice, GABAergic synaptic transmission is markedly impaired from birth [124,125], influencing MN numbers, morphology and NMJ innervations [124,126], with synaptic defects occurring later in GAD65 knockouts [127,128]. By contrast to GABA synthesis, deficiency in glycine synthesis (a serine hydroxymethyltransferase null mutant) does not seem to have many overt effects [129].

VGAT is highly influential on in utero development, with marked cleft palate, omphalocele and abnormal spine curvature, of a greater extent than those of GAD mutants [117,123,130,131]. These gross developmental morphology defects are likely due to altered foetal movements [117,118,125], in a similar manner to how lung development is impaired when foetal diaphragm muscle or breathing activity is reduced [117,118,132]. In support, despite excess GABA and glycine (i.e., an abundance of “trophic” inhibitory amino acid signalling) in VGAT mice, there is an almost total absence of GABAergic and glycinergic neurotransmission [124,125], with marked effects on MN survival, morphology and muscular innervation [124,133].

GABA receptor subunit knockout mutations have explored the role of GABA-A receptor-mediated synaptic influences on morphology, with β_3_, during embryonic development leading to cleft palate deformities [134,135], while γ_2_ exhibits a postnatal developmental growth retardation and sensorimotor dysfunction [136]. However, as many of the major α and β subunits exhibit differential expression following postnatal development, in utero effects are not as common as in most other mutations [137]. With regard to the glycine receptors, there are some human hyperekplexia conditions directly tied to mutations in α_1_ glycine receptor subunits [138,139]. Knockouts of the α_2_ glycine receptors do not exhibit gross morphological deformities or growth restrictions and following the establishment of mature glycine signalling synaptic differences are absent, indicating compensation by other subunits [138,140]. Similarly, knockouts of glycine receptors do not exhibit gross morphological alterations [138,141].

Altered synaptic stabilization molecules are noted for their influence on development. At GABAergic synapses, Neuroligin-2 is predominantly associated with neurobehavioural disorders, including autism and attention deficit hyperactivity disorder and reduced GABAergic inhibitory neurotransmission [142,143,144]. By contrast, the gephyrin mutant, which lacks the glycine receptor binding and destabilization has a phenotype at birth, with marked startle responses and death from respiratory muscle dysfunction [145,146]. In gephyrin deficient mice, there is an almost total absence of glycinergic neurotransmission, which is not compensated for by GABAergic transmission (which remains functional) [17,133,147] alongside severe effects on MN survival, morphology and muscular innervation [133]. These results are also suggestive of different postsynaptic roles for gephyrin at GABAergic and glycinergic synapses, which may also depend on neural region [148,149] and pathophysiological factors [150]. Additional roles for gephyrin impairment in various behavioural brain disorders have also been postulated [151].

Altered ligand reuptake by GABA transporters (GAT1) for GABA, and ASC1 for glycine have impaired inhibitory neurotransmission, with the motor phenotype more pronounced in GlyT2 and ASC1 mutants than in the GABA reuptake mutants [152,153,154,155,156,157]. Notably, the severity of the phenotypes of many of these reuptake mutants is dependent on developmental timing, with different transporter expression and differential salience of GABA and glycinergic neurotransmission at different ages [71,105,158,159].

### 2.4. Developmental Aspects of Chloride Channel Expression

In utero, and during early postnatal development the ionotropic neurotransmitter actions of GABA and glycine are depolarizing, rather than hyperpolarizing [160,161]. This is due to the differential regulation of chloride cotransporters. In immature neurons, the expression of Na^+^-K^+^-Cl^−^ cotransporter 1 (NKCC1) is elevated compared to K^+^-Cl^−^ cotransporter 2 (KCC2) [161,162]. NKCC1 increased intracellular chloride, thus when the ligand binding opens the chloride pore at GABAergic and glycinergic synapse, chloride efflux occurs, depolarizing the neuron [163] (Figure 3). By contrast, in mature neurons, KCC2 transporters are highly expressed, with a relatively low intracellular chloride concentration resulting in the influx of chloride following ligand binding at GABAergic and glycinergic synapses and neuronal hyperpolarisation [163,164] (Figure 3).

The temporal and spatial nature of NKCC1 to KCC2 maturation has been a topic of intense interest and may be different between GABAergic and glycinergic systems [112,160,165]. For GABA, maturation seems to occur in a caudo-rostral fashion, with maturation occurring first in the spinal cord [166], followed by the brainstem [167], cerebellum [168], hippocampus [169,170] and then the cortex [171]. At glycinergic synapses, maturation seems to occur at similar times throughout the neuraxis [112,160,165,166,167,172,173,174].

Intriguingly, from a pathophysiological perspective, a variety of conditions alter the expression of these chloride transporters, including Alzheimer’s disease [175,176], anaesthesia [177], autism [175,178,179], epilepsy [167,175,180,181,182,183], Huntington’s disease [175,184,185,186,187], Rett syndrome [178,188], schizophrenia [175,189,190] and traumatic brain and spinal cord injury [181,191,192,193,194,195].

In addition to chloride, GABAergic receptor channels are permeable to bicarbonate (HCO_3_^−^), generated as part of the cellular carbon dioxide (CO_2_) conversion [196]. When the GABA ionic pore is open, this causes bicarbonate efflux, depolarizing the neuron [197]. Under standard conditions, the permeability of chloride is ~4–5 times greater than that of bicarbonate [198]. Under certain conditions, such as seizures or when there is an excessive period of chloride permeable channel opening, there may be a dissipation [199,200] or reversal [201] of the chloride gradient, leading to depolarization of the neuron due to the bicarbonate efflux [197,202]. In the context of many neurodevelopmental disorders, bicarbonate action is understudied and may be a fruitful direction of investigation in conditions of intractable excitation/inhibition imbalance.

## 3. Inhibitory Influences on Rett Syndrome

### 3.1. Rett Syndrome

Rett syndrome is an early-onset neurological disorder that commences between 6–18 months of age [203,204]. These children exhibit typical neural motor and language development until this period, with the stark arrest of typical developmental milestones and the onset of symptoms including hand-stereotypies, breathing and gait abnormalities coinciding with growth reduction—particularly of the head [203]. By 4–7 years old, children with Rett syndrome have motor and cognitive deficits that will continue throughout their lifespan [203] in addition to a variety of neurobehavioural issues consistent with autism [203]. There is a direct genetic link for Rett syndrome, a mutation in the X-linked *MECP2* gene [205], with females overwhelmingly more likely to be affected [203,206]. The *MECP2* gene is highly expressed in neurons and is essential for neuronal development and in the establishment of synaptic connections [207], with normal function involving both repression (via binding to methylated DNA) or activation of gene transcription [208]. Mutated *MECP2* results in impaired *MECP2* protein synthesis and a lack of repression/promotion activities. The developmental onset of *MECP2* expression during the postnatal period is contemporaneous with central nervous system maturation [209,210], and closely matched the timing of initial Rett syndrome symptoms [207]. However, in Rett syndrome there are many individual *MECP2* mutations, neither necessary nor sufficient for diagnosis [211,212]. Males that are affected develop the syndrome extremely rapidly (within a few days postpartum) and die within 2 years [206,213]. Recently, there have been major advancements in the amelioration of Rett syndrome via growth factors (IGF1), *MECP2* gene transfer, genetic corrections and approaches focusing on downstream targets in rodent models and human trials [207,214,215,216,217,218,219]. Despite these promising advances targeting *MECP2* and its related downstream protein, there remains an influence on Rett syndrome symptoms of altered inhibitory neurotransmission on the motor phenotypes and altered chloride homeostasis [220]. A richer understanding of the involvement of inhibitory neurotransmission and altered chloride homeostasis may pave the way for alternative therapeutic interventions in cases where genetic approaches are precluded or the nature of specific individual patient mutations are unknown.

### 3.2. Limb and Respiratory Neuromotor Deficits in Rett Syndrome and Their Relationships to Inhibitory Neurotransmission

As noted above, even in a simplified context, there are many mediators of motor control prior to their convergence on the motor unit as the final common pathway (Figure 1). In addition, much of respiratory neuromotor control is misattributed to the autonomic nervous system. Insights into the specific movement dysfunctions of the limb, respiratory and respiratory associated (including speech, cough [1,221,222]) neuromotor behaviours may uncover overlooked aspects of inhibitory influences on Rett syndrome. Indeed, in Rett syndrome patients and in *MECP2* models, reduced GABA-A receptor expression is evident [223,224], although excitation/inhibition imbalances appear to be region specific [220]. Curiously, mutation of *MECP2* in exclusively VGAT-expressing cells (i.e., inhibitory interneurons—see above), recapitulates the Rett syndrome phenotype [225]. When *MECP2*-deficient mice have *MECP2* rescued in exclusively VGAT-expressing cells a substantial improvement of Rett syndrome related symptoms ensues [226]. In patient neurons derived from induced pleuripotent stem cells, GABAergic dysfunction is also evident, further underscoring the importance of inhibitory deficits in Rett syndrome [225].

Limb associated motor dysfunctions may be under greater influence by corticospinal pathways than those of respiratory or respiratory associated motor dysfunctions [10] (Figure 1). Excessive inhibition in the motor cortex in *MECP2* models [216,227,228] is not found in patients, which exhibit cortical hyperexcitability [229,230]. Despite there being difficulties in localizing specific pathophysiologies of motor stereotypies, reduced GABA in cingulate and striatal regions were associated with a patient population with complex motor stereotypies [231]. Motor stereotypies in mice have been associated with altered Neuroligin-2 expression, further underscoring the importance of inhibitory neurotransmission and the motor dysfunctions [232]. Overall, despite some differences between *MECP2* models and the human scenario, therapeutic approaches involving inhibitory modulation may prove beneficial in the limb dysfunctions observed in Rett syndrome.

The potent influence of inhibitory neurotransmission underlies many aspects of the rhythmic nature of breathing [72,99,233,234,235,236] and the coordination of breathing with other activities [2,10,237]. Considerable attention has been paid to respiratory neuromotor dysfunction in Rett syndrome, with breathing disturbances, alongside coordination of breathing problems (i.e., swallowing, speech, etcetera) common across patients and *MECP2* models [37,203,238,239,240,241,242,243,244,245,246]. Importantly, breathing dysfunctions occur during periods of wakefulness and sleep, with cessation of breathing during awake states not necessarily apnoeas [37,247,248], but prolonged breath-holds [37,240,246,248] that more closely resembling Valsalva manoeuvres [37]. By contrast, during sleep, obstructive sleep apnoeas are more prevalent [245,249,250,251,252,253,254]. It is not currently known as to whether the intensity of the Valsalva-like activation of the diaphragm muscle mirrors the maximum magnitudes of pressure generation observed in adults [37], which exhibit transdiaphragmatic pressures ~10 times that of eupnoea [10,255,256,257], but if it is of sufficient magnitude, it may indicate that activation of type S, type FR and type FF motor units is altered in an arousal- dependent manner in Rett syndrome. Studies in *MECP2* mice indicate that these breathing abnormalities may be related to post-inspiratory phase of breathing via altering the activity of the Kölliker-Fuse nucleus, the nucleus tractus solitarius and the ventrolateral medulla (see [37] for an excellent summary of the *MECP2* discoveries of breathing circuit abnormalities) and reduced inhibitory inputs to various respiratory associated brainstem regions [220,258,259,260,261]. Of particular relevance to inhibitory pathways, a discoordinated relationship between hypoglossal MNs and the post-inspiratory phase and disinhibition of related circuits is observed in *MECP2* models [262,263] and is consistent with altered speech and swallowing control [264] in Rett syndrome [37,265,266,267]. Improvement of GABAergic signalling in *MECP2* mice appears to restore normal breathing [258,268,269].

Alongside the major limitations for speech when discoordination of the lingual and oropharyngeal muscles and the post-inspiratory phase occur, dysphagia and aspiration present major health concerns for Rett syndrome patients [267,270,271]. Indeed, pneumonias are the primary morbidity and mortality indicated in Rett [271,272,273]. Rett syndrome is not coupled to ineffective airway clearance reflexes [267], although no evaluation of airway clearance pressures has been performed.

In summary, despite clear evidence of inhibitory influences on the pathophysiology of Rett neuromotor dysfunctions, specific interventions remain few and far between. With time, evaluation of a variety of GABAergic and glycinergic agents aimed at improving the regularity of respiratory and oromotor associated neuromotor circuit discharge (Figure 1) may yield major advances for patient quality of life.

### 3.3. Chloride Homeostasis in Rett Syndrome

Due to the developmental nature and the large degree of symptomatic heterogeneity observed in Rett syndrome, various mechanisms of spatio-temporal governing of neural circuit function have been proposed as disease-influencing. One such mechanism is the shift in neuronal intracellular chloride concentration from high to low via the mature expression of KCC2. Notably, KCC2-deficient mice exhibit similar pathologies of breathing abnormalities, lower body mass and impaired cognition and memory to those of *MECP2* models [274,275,276]. In Rett syndrome, KCC2 levels are altered in patients, neurons derived from patient induced pluripotent stem cells and the *MECP*2 model [178,188,277,278,279], with suggestions that KCC2 is downstream of *MECP2* [280], or reduced *MECP2*-influenced BDNF signalling [281,282,283] suppresses KCC2 [284], which is reduced in patient neurons derived from induced pluripotent stem cells [278]. Rescuing or enhancing KCC2 in *MECP2* mice or patient induced pluripotent stem cells using a variety of treatments has been shown to be a feasible approach [285,286]. Overall, aside from the major genetic contribution (~80%) of *MECP2* to Rett Syndrome, patients with and without the mutation have a reduction in the abundance and perhaps the maturation of inhibitory neurotransmission, independent and dependent of *MECP2*.

## 4. Inhibitory Influences on Spastic Cerebral Palsy

### 4.1. Spastic Cerebral Palsy

Cerebral palsy is the most common childhood motor disability [287,288,289,290], with ~80% of clinical presentations involving spasticity (sCP) [38,290,291]. Despite the aetiology of sCP being a topic of intense debate [38,292,293,294,295,296,297,298,299], the symptoms, therapeutic interventions and the timing of syndrome onset is consistent with a major influence of inhibitory neurotransmission on sCP [300,301]. Symptomatically, cerebral palsy is characterized by spasticity (sCP), dyskinesias, ataxia, hyper-reflexia and occasionally hypotonia [290,302,303]. Notably, spasticity and hypertonia, defined clinically as the resistance of a muscle to stretch [304,305,306]. and hyper-reflexia, increased stretch reflex responses, are thought to involve specific disinhibitory mechanisms [300,301,307,308]. Moreover, the timing of the development of the sCP syndrome is concomitant with the postnatal maturation of chloride channels (NKCC1 to KCC2—see above) throughout the neuraxis [163,309,310]. Motor control is hindered in limb muscles [38,311], with the respiratory muscles and speech often affected due to either weakness or post-inspiratory motor control problems [312,313]. Despite the empirical evidence of inhibitory neurotransmission being a potential key player in the development and symptoms of sCP, an implacable focus of many preclinical studies has remained on perinatal injury and hypoxia, which are neither predictive nor specific for sCP [38,295,299,314]. Nonetheless, an emerging body of evidence in a variety of animal models, with some supporting clues from clinical investigations and therapeutic interventions has uncovered an involvement for inhibitory neurotransmission defects within the spinal cord (and upon MNs) in sCP.

### 4.2. Animal Models of Cerebral Palsy and Criteria for Validation

By contrast to the highly successful *MECP2* model for Rett syndrome, models of cerebral palsy in general, and sCP in particular, are not widely adopted and most have little resemblance to the clinical scenario. Importantly, a diversity of clinical presentations in Rett syndrome has not hindered the adoption of the *MECP2* model, while the diversity of sCP presentations seems to have spurred a multitude of animal models, all with different goals—from modelling risk factors to modelling symptoms.

As mentioned, models of cerebral palsy fall into two broad categories, namely, models of risk factors and models of symptoms. Some mooted risk factors include perinatal anoxia, perinatal brain injury, perinatal infection, congenital deformity and genetic factors [38,296,315]. However, some of these risk factors are neither sensitive nor specific to cerebral palsy, for example, brain injury may be present without cerebral palsy, and cerebral palsy may be present without brain injury per se, perhaps due to lack of uniform assessment standards [295,299,316]. Indeed, all the known risk factors besides birth prematurity (with ~75% of cerebral palsy patients having been born premature), display rates of cerebral palsy in <15% of all cases (in utero ischemia/infection/inflammation 12%, stroke < 1%, hypoxia-ischemia encephalopathy 15% and asphyxia 2–10%) [38]. Nonetheless, hypoxia, ischemia, in utero infections and combinations of these have been modelled in rabbits, rats and many other species [38,315,317]. Unsurprisingly, due to the low risk-association with cerebral palsy, most of these models do not recapitulate motor dysfunctions [38,318], spasticity or hypertona [38,318,319], have an excessive mortality rate or deaths shortly following experiments (unlike the clinical scenario, where mortality in associated with symptom severity) [292,319,320,321] and lack spasticity [38,292,318]. One exception to this general lack of suitability is the rat model of transient occlusion of uterine arteries and intra-amniotic injection of lipopolysaccharide [317,322], which despite high experimental mortality [317], the surviving offspring exhibit locomotor and postural deficits and some indications of spasticity [317], although there seems to be ventriculomegaly, other white and grey matter lesions and cognitive declines that do not match some aspects of clinical sCP [317]. Overall, the risk factor models succeed in providing information about the impact of various lesions on brain development but do little to move the needle of sCP translation.

One model of sCP that has been guided by the natural history of the clinical syndrome is the *Spa* mutant mouse. The *Spa* mice have a mutation in the β subunit of the glycine receptor gene resulting in a splicing error prevalent in the brain and spinal cord [323]. *Spa* mice develop spasticity and hypertonia ~2–4 weeks postnatal [319,323,324], have motor deficits [256,325,326,327,328,329] and restricted growth [319], with mild to moderate mortality rates [319]. Despite this surface suitability, the majority of the sCP research community seems fixated on in utero surgical approaches modelling the poorly defined “brain injury” that has captured much of the clinical imagination [292].

If one considers the major tenets of an animal model to exhibit: (i) the behavioural phenotype; (ii) the pathological phenotype; (iii) the temporal phenotype and (iv) the aetiological mechanism of a particular disease, the *Spa* mouse exemplifies sCP in points i, ii and iii. Despite this, there is no major association of glycinergic mutations with cerebral palsy, point iv. However, if we expand our definition of aetiology to include disease manipulating mechanisms, then the *Spa* model is clearly one of the better options in our arsenal. Of course, an animal model based on a known risk factor that recapitulates tenets i–iii would be a major breakthrough in sCP research. The remainder of this section of the review will focus on the importance of inhibition, specifically inhibition and its influence on the neuromotor system and symptomatic treatment in clinical sCP and in experimental model approaches.

### 4.3. Evidence for Inhibitory Deficits in Spastic Cerebral Palsy

As mentioned previously, inhibitory neurotransmission dysfunction in developmental behavioural and psychiatric disorders is well-established [36]. Clinical studies and animal models have shown that dysfunction may involve reduced levels of inhibitory neurotransmitters in cerebro-spinal fluid [330,331], the GABA and/or glycine receptors (reduced expression levels or loss-of-function mutations) [332,333,334], reduced inhibitory circuit inputs [335,336,337] and altered chloride channel expression [338,339,340,341]. In cerebral palsy, there is evidence of decreased GABA-A receptors in the cerebrospinal fluid [342] and reduced GABA receptor expression in the brainstem, but not the cortex (a further nail in the coffin of the cerebral primacy of the aetiology) [343], deficits in reciprocal inhibition [344,345,346], suggestions of reduced cortical inhibition [347,348] and a relationship between the of loss of inhibition and the degree of physical motor dysfunction [349]. Together, these scraps of information in sCP patients and in spasticity more generally led to the trial of a various strategies to enhance or compensate for inhibitory dysfunctions.

### 4.4. Enhancing Inhibition as Therapies in Spastic Cerebral Palsy

Response to treatment has historically been important in the de facto understanding the pathophysiology of many neurological defects. Indeed, in sCP, a multitude of treatments aimed at improving inhibitory neurotransmission or compensating for reduced inhibitory neurotransmission, provide some ersatz mechanistic understanding of the syndrome. These treatments that hint at inhibitory involvement include intrathecal baclofen and benzodiazepines [290,350]. Intrathecal baclofen, a GABA-B receptor agonist [351,352] and mediate some GABA-A activation [353,354] has been used to treat spasticity in general [355] and sCP [356] since the 1980s. It has been shown to reduce spasticity and improve the quality of life of sCP patients, both children and adults, in a variety of trials an meta analyses [357,358], although hypotonicity in unaffected limbs remains a concern [358]. Curiously, despite the recent use of *Spa* mice with a glycine receptor mutation closely mirroring the natural history of sCP [256,292,319,329,359], glycine therapy (which crosses the blood–brain barrier) has not been more widely adopted, despite some promising preliminary studies in sCP patients and other spastic conditions [360].

Benzodiazepines (most notably diazepam) bind to GABA-A receptors [361] and reduce muscle spasticity via inhibiting MNs in patients with sCP [350]. Clinically, improvement post diazepam treatment has been demonstrated [362,363,364], with nocturnal dosing helping to avoid daytime sedation [364]. Longer-term use of diazepam is somewhat hindered by a building up of tolerance and consequent dosage escalations [365], severe withdrawal effects [365] and concerns regarding dependence and abuse [365,366].

Importantly, the efficacy (although modest in many cases [290]) of these interventions can be interpreted as being consistent with sCP being a disorder of inhibition. Secondly, and perhaps most importantly, the underlying pathophysiology of sCP may reside in the impairments of inhibitory neurotransmission throughout the motor neuraxis; and are valid regardless of the prism through which one views the aetiology of sCP (i.e., brain injury, hypoxia/ischemia and maternal age).

### 4.5. Motor Neurons and the Spinal Cord as a Locus of Pathophysiology in Spastic Cerebral Palsy

Other clues to the importance of the spinal cord in sCP are in the original description as a cerebro-spinal syndrome (named Little’s disease) [367]. Indeed, the inclusion of the spinal cord is evident by descriptions of MN degeneration in patient autopsy [368], reduced motor unit number estimates [369] and a reduction in the grey and white matter of the spinal cord [290,293,370,371]. Notably, in the *Spa* model of early onset spasticity, MN loss is prevalent in the spinal cord, disproportionately afflicting the larger MNs of the phrenic and tibialis anterior motor pools [325,326,327]. In the rabbit ischemia/hypoxia model, generalized lumbar MN loss was evident, without equivalent loss in the cervical spinal cord [372] was observed perinatally. In this particular study, no attempt was made to determine the size-dependence of loss. Regardless, in leporidae MN size in the first postnatal week is not indicative of motor unit types, as motor unit type dependent expansion of MN and muscle fibre size, in addition to the expression of the full gamut of mature myosin heavy chain isoforms occurs later in development in rodents (which leporidae were once classified) [373,374,375,376] and leporidae themselves [377,378,379,380]. Interestingly, reducing excitatory AMPA receptor activation in this model restores some function in rabbit ischemia-hypoxia models [381].

The importance of the MN in sCP is not solely related to the rostro-caudal neuroanatomical locus of pathology. As the final common pathway in the motor system, MNs may serve as the ideal site for therapeutic interventions. Indeed, four major therapeutic interventions baclofen injection, dorsal rhizotomy, botulinum toxin type A (BoNTA) and dantrolene are localized to the spinal cord (spinal baclofen and dorsal rhizotomy) and/or motor unit (baclofen and dorsal rhizotomy affecting MNs directly or indirectly, with BoNTA at the NMJ and dantrolene at the skeletal muscle). Perhaps the central nature of the motor unit to the symptoms of sCP is evident in the use of BoNTA as a therapeutic to reduce spasticity in spinal cord injury, stroke and sCP [290,382,383]. Although BoNTA does not influence the inhibition of MNs, it does make use of the motor unit as a final common pathway by blocking the acetylcholine release at the NMJ. Although widely used for sCP [290,383,384,385], the nature of any developmental effects (on MNs and muscle) that decoupling MN firing from skeletal muscle excitation–contraction coupling is a concern [383], albeit one with a very little mechanistic valuation. In addition, whether repeated dosing of BoNTA leads to exaggerated muscle weakness and atrophy is a source of controversy [383]. Regardless, studies in animals are not performed under conditions of MN loss, nor at stages where developmental myosin is still being expressed in muscle, which represents a major knowledge gap. Dantrolene directly limits skeletal muscle contractility by inhibiting excitation–contraction coupling [351,386] and is advised for those whom centrally acting agents or general sedation may be contraindicated [350]. Improvements in reduced hypertonicity, range of motion, strength (likely due to reduced joint torque) and spasticity are noted [350,387], although outcomes and side-effects, including longer-term weakness remain understudied [290,388].

### 4.6. Prospects for Progress in Understanding the Pathogenesis and Improving Patient Outcomes in Spastic Cerebral Palsy

Despite cerebral palsy being the most common early onset movement disorder, there has been a remarkable lack of innovation in the therapeutic arena—likely due to the preponderance of the (little) research attention paid to cerebral palsy fixated on modelling perinatal risk factors rather than symptoms. Some attention paid to the pathophysiology of weakness and the roles MN loss or dysfunction may play in the development and progression of sCP is long overdue.

Intriguingly, a size-dependent loss of MNs hints at motor unit specific effects on muscle contractility and fatigue with type FF motor units likely to bear the major phenotypic and symptomatic burden, with type S and FR motor units relatively preserved. This contention holds in the *Spa* mice, where neuromuscular transmission failures and behavioural assessments show deficits or overt weakness, reflective of the action and/or recruitment of FF motor units [256,327,328,329]. These findings are consistent with some human investigations, where overt muscle weakness is readily demonstrated [389,390,391], particularly with increased symptom severity [392] with a conservation of muscular fatigue and endurance properties [390,393,394]. Notably, the integrated muscular function during activities such as walking may be impaired in cerebral palsy due to constant co-activation of antagonistic muscles and/or cardiovascular or respiratory factors [312,391,394,395].

Currently, a “cure” for sCP remains a pipe dream, fuelled by desperation and thwarted by both the heterogeneous nature of the syndrome and perhaps more importantly by ethical considerations amongst the adult398 cerebral palsy community that (rightly) demand a focus in improving their quality of life (see [396,397] for a detailed discussion on this topic). By contrast, a resetting of some research priorities towards maximizing the efficacy and safety of existing therapies (with two major interventions directly involving inhibitory inputs to MNs) or developing uncovering new targets or pharmacological agents that improve inhibitory neurotransmission or neural circuitry within the brain and spinal cord is well past due. Finally, in the absence of a widely adopted and validated (i.e., symptomatic) model of sCP that recapitulates the natural history of the syndrome, we remain incredibly limited in our abilities to rationally test any new interventions. Until we have the creation of a new model or the adoption of the “least-worst” symptomatic proxy (perhaps the *Spa* mouse) any new therapy will be untested and captive to the (quite rightly) risk averse proclivities of clinicians. These approaches we suggest here are agnostic as to disease aetiology and serve to improve the lives of those currently living with cerebral palsy as well as those yet to be born.

## 5. Conclusions

The abundance of transporter and receptor subtypes of the structures involved in GABAergic and glycinergic neurotransmission should be seen by researchers as an opportunity as opposed to a drawback. The diversity of isoforms, and differences in ligand affinities may prove to multiply the potential therapeutic approaches and targets designed to arrest excitation/inhibition imbalance and disorders of inhibition. In two developmental neuromotor disease contexts, Rett and sCP, intuitive approaches to limiting disinhibition or more mechanistic approaches aimed at restoring the correct maturational factor involved in inhibitory neurotransmission show promise. Despite these advances, in sCP symptomatic therapies remain a matter of guesswork without a widely adopted and validated symptomatic model. By contrast, great strides in Rett syndrome have been made despite the imperfections of the *MECP2* model, due to the fields focus on the restoration of a clear clinically important phenotype—namely breathing. It is this author’s opinion that the clarity that we have regarding the circuit control of the respiratory neuromotor system provides a stable backdrop for the evaluation of the pathophysiology and potential therapies for a variety of developmental and degenerative neuromotor diseases.

## Figures and Tables

**Figure 1 ijms-24-06962-f001:**
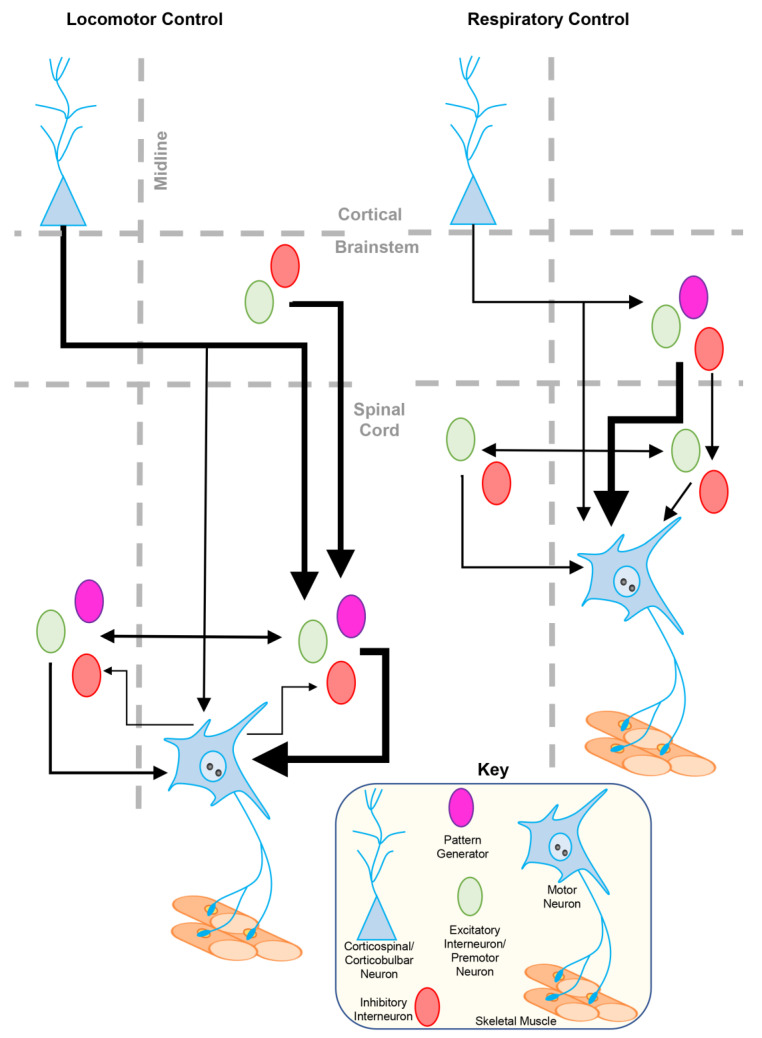
Locomotor and respiratory neuromotor control. In the case of locomotor control, there are corticospinal influences (blue pyramidal neurons) and bulbospinal influences on spinal cord central pattern generators (purple), and inhibitory (red) and excitatory (green) premotor neurons. These segment-level neurons coordinate motor outputs of the ipsilateral and contralateral side via projections. Ipsilateral and contralateral inhibition via circuits formed from recurrent motor axons also provides for coordination within these segments. In the locomotor system, the major inputs to motor neurons (blue neurons innervating skeletal muscle fibres) come from ipsilateral pattern and premotor neurons within the spinal cord segment. For respiratory neuromotor control, pattern generation and premotor excitatory and inhibitory circuits reside primarily in the brainstem, projecting to ipsilateral phrenic motor neurons. There are some segment-level premotor inputs to motor neurons; however, all pattern generation necessary for breathing resides in the brainstem, with a modicum of cortical modulation for a variety of post-inhibitory behaviours (e.g., speech/vocalisation).

**Figure 2 ijms-24-06962-f002:**
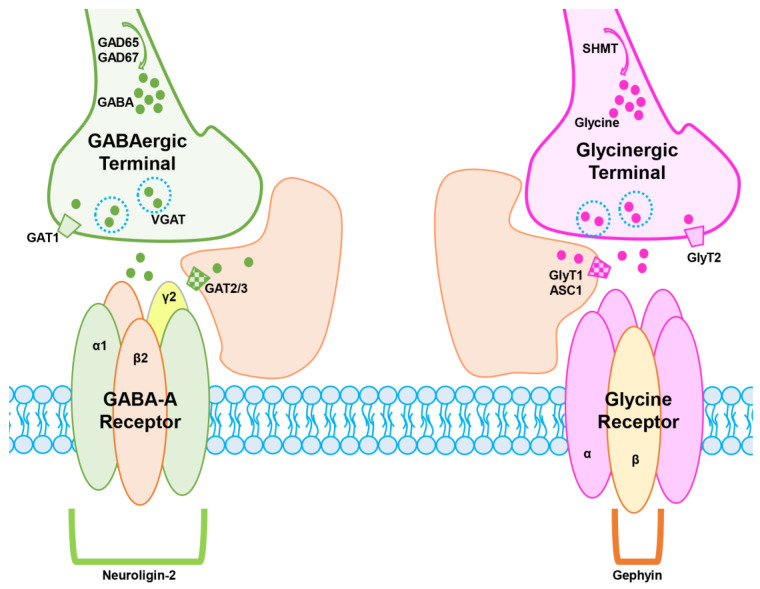
GABAergic and glycinergic synapses. GABA (green circles) is synthesized by GAD65 and GAD67 (green arrow) and packaged into synaptic vesicles by VGAT (blue dashed circles). Upon release into the synaptic cleft, GABA is bound by αβ subunit binding sites (green and orange subunits) at GABA-A receptors. γ subunits (yellow) provide the binding sites for modulators such as benzodiazepines. Following release into the synaptic cleft, GABA reuptake occurs presynaptically via GAT1 (green rhomboid) and in astrocytes (orange) via GAT2/3 (checked green rhomboid). GABA-A receptors are structurally maintained by Neuroligin-2 (green bracket). Glycine (pink circles) is synthesized by serine hydroxymethyltransferase (SHMT—pink arrow) and packaged into synaptic vesicles by VGAT. Upon release into the synaptic cleft glycine is bound by α subunit binding sites (pink subunits) at glycine receptors. Following release into the synaptic cleft, Glycine reuptake occurs presynaptically via GlyT2 (pink rhomboid) and in astrocytes via GlyT1/ASC1 (checked pink rhomboid). β glycine receptor subunits (peach) provide the binding sites for gephyrin (orange bracket) in order to structurally maintain the synapse.

**Figure 3 ijms-24-06962-f003:**
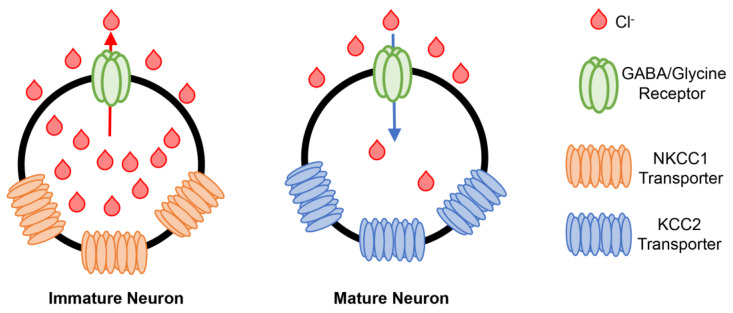
Maturation of neuronal chloride transporters. In immature neurons, chloride (Cl^−^, red drops) concentrations are higher intracellularly due to the expression of immature NKCC2 transporter (orange). When the ion pore of GABAergic or glycinergic synapses is opened (green), chloride efflux (red arrow) occurs depolarizing the neuron. In mature neurons, intracellular chloride concentration is low, due to the expression of the mature KCC2 transporter (blue). When the ion pore of GABAergic of glycinergic synapses is opened chloride influx occurs (blue arrow), hyperpolarizing the neuron.

## Data Availability

Not applicable.

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
