# Peer review of "Inhibitory Synaptic Influences on Developmental Motor Disorders"

_ijms, 2023, doi:10.3390/ijms24086962_

Round 1

Reviewer 1 Report

The review article entitled “Inhibitory synaptic influences on developmental motor disorders” by Matthew J. Fogarty describes the issue of inhibitory synaptic transmission in the pathophysiology of neuromotor circuits. The main goal of the Author is a comprehensive description of the role played by GABAergic and glycinergic neurotransmission in two neurodevelopmental disorders: cerebral palsy and Rett syndrome. Contemporary neuroscience explores the mechanisms that keep the balance (local and global) between synaptic excitation and inhibition. It is well established that changes in this balance underlie normal and pathological brain development. It is worth noting that the manuscript extends this knowledge into the analysis of spinal cord circuits and the neuromotor unit.

In particular, abnormal changes in chloride balance may underlie many neurodevelopmental disorders such as Autism Spectrum Disorder and Rett Syndrome. Thus, a pharmacological intervention aimed at normalizing the chloride balance in neurons presents an interesting and promising option in the development of new therapies for the above-mentioned disorders. Importantly, preliminary clinical data support this line of research.

The review is, especially with provided figures, very informative and helps its readers to understand a thorough picture. This review/perspective would be suitable for researchers of inhibitory transmission in the brain but also in the clinical fields.

I have only couple minor comments as listed below:

1.      Line 83-89, please provide necessary details on the regions in the brain (or spinal cord) from which the cited numbers were obtained;

2.      Line 91-95 “Direct GABAergic inhibition is particularly important in the more cranial regions of the central nervous system, such as the cortex, hippocampus, thalamus and brainstem [55,57,69], whereas presynaptic GABAergic synaptic inhibition may play more of a role in MNs, particularly in the lumbar spinal cord [71-74].” This sentence is not clear for me. Please explain the difference between “direct GABAergic inhibition” and “presynaptic GABAergic synaptic inhibition”;

3.      Line 104-105 “Each subunit exhibits splice variants [77], however, the majority of synaptic GABA-A receptors are α1β2γ1” Again, it depends on the brain region, please specify;

4.      Line 106. GABAARs with alpha5 subunit contribute also substantially to tonic inhibition (e.g. PMID: 33096025);

5.      Line 128. The sentence “which is predominantly in the brainstem and spinal cord” repeats what has been already mentioned in the previous sentence of this paragraph;

6.      Line 212-213. The cited results are very important because they show that despite the corelease of GABA and glutamate, the lack of gephyrin mainly affects glycinergic transmission. This may suggest that the different protein composition of the postsynaptic density of both inhibitory synapses shapes a different role of the key anchoring protein, gephyrin. Please consider adding one sentence summarizing this effect;

7.      Lines 224-233. Please consider mentioning that HCO3- ions contribute as well to synaptic GABAARs currents. Moreover, what is important, changes in HCO3- concentration are understudied in the context of brain development or neuropathologies.

8.      Line 288-291: “Although the precise mode of action of many gaseous and intravenous anaesthetics remains unknown, in the case of isoflurane, potentiated GABA-A and glycine receptor signalling and with propofol GABA-A receptors [234]” The sentence is unclear. Please reformulate;

9.      Page 11. Figure 1 with caption appear for the second time in the manuscript;

10.  Line 404. Please consider adding few details about MECP2 protein and its roles. In the rest of this chapter the gene and protein MECP2 are mentioned numerous times without any details about its function.

Author Response

REVIEWER 1

General Comment: The review article entitled “Inhibitory synaptic influences on developmental motor disorders” by Matthew J. Fogarty describes the issue of inhibitory synaptic transmission in the pathophysiology of neuromotor circuits. The main goal of the Author is a comprehensive description of the role played by GABAergic and glycinergic neurotransmission in two neurodevelopmental disorders: cerebral palsy and Rett syndrome. Contemporary neuroscience explores the mechanisms that keep the balance (local and global) between synaptic excitation and inhibition. It is well established that changes in this balance underlie normal and pathological brain development. It is worth noting that the manuscript extends this knowledge into the analysis of spinal cord circuits and the neuromotor unit.

In particular, abnormal changes in chloride balance may underlie many neurodevelopmental disorders such as Autism Spectrum Disorder and Rett Syndrome. Thus, a pharmacological intervention aimed at normalizing the chloride balance in neurons presents an interesting and promising option in the development of new therapies for the above-mentioned disorders. Importantly, preliminary clinical data support this line of research.

The review is, especially with provided figures, very informative and helps its readers to understand a thorough picture. This review/perspective would be suitable for researchers of inhibitory transmission in the brain but also in the clinical fields.

Response: We thank the reviewer for their encouraging response.

Minor Comments

  1. Line 83-89, please provide necessary details on the regions in the brain (or spinal cord) from which the cited numbers were obtained;

Response: We have indicated these details in the revised manuscript.

  1. Line 91-95 “Direct GABAergic inhibitionis particularly important in the more cranial regions of the central nervous system, such as the cortex, hippocampus, thalamus and brainstem [55,57,69], whereas presynaptic GABAergic synaptic inhibition may play more of a role in MNs, particularly in the lumbar spinal cord [71-74].” This sentence is not clear for me. Please explain the difference between “direct GABAergic inhibition” and “presynaptic GABAergic synaptic inhibition”;

Response: We have clarified the effect of direct referring to neuronal hyperpolarization and “presynaptic” referring to inhibition at axon terminals.

  1. Line 104-105 “Each subunit exhibits splice variants [77], however, the majority of synaptic GABA-A receptors are α1β2γ1” Again, it depends on the brain region, please specify;

Response: We have clarified the regions and noted the variability that is likely apparent in development and across regions.

  1. Line 106. GABAARs with alpha5 subunit contribute also substantially to tonic inhibition (e.g. PMID: 33096025);

Response: We thank the reviewer for drawing this study to our attention and have briefly included a point about the dynamics of a5 subunits.

  1. Line 128. The sentence “which is predominantly in the brainstem and spinal cord” repeats what has been already mentioned in the previous sentence of this paragraph;

Response: We have clarified this section.

  1. Line 212-213. The cited results are very important because they show that despite the corelease of GABA and glutamate, the lack of gephyrin mainly affects glycinergic transmission. This may suggest that the different protein composition of the postsynaptic density of both inhibitory synapses shapes a different role of the key anchoring protein, gephyrin. Please consider adding one sentence summarizing this effect; Response: We have added a comment to this effect.
  2. Lines 224-233. Please consider mentioning that HCO3- ions contribute as well to synaptic GABAARs currents. Moreover, what is important, changes in HCO3- concentration are understudied in the context of brain development or neuropathologies.

Response: We thank the reviewer for suggesting this addition and have includes a small  section on bicarbonate and depolarization.

  1. Line 288-291: “Although the precise mode of action of many gaseous and intravenous anaesthetics remains unknown, in the case of isoflurane, potentiated GABA-A and glycine receptor signalling and with propofol GABA-A receptors [234]” The sentence is unclear. Please reformulate;

Response: We have removed this section as its clinical justifications in CP particularly are relatively weak.

  1. Page 11. Figure 1 with caption appear for the second time in the manuscript;

Response: This has been corrected.

  1. Line 404. Please consider adding few details about MECP2 protein and its roles. In the rest of this chapter the gene and protein MECP2 are mentioned numerous times without any details about its function. Response: We have added a small section on MECP2.

Reviewer 2 Report

This review approaches the interesting topic of inhibitory synaptic influences focusing on developmental motor disorders. The conceptualization and generous references lead to an attractive paper. However, it would be necessary to resolve some issues to improve its quality.

Major issues:

-       There is a gap between the second part (2. Inhibitory neurotransmission) that is full of detailed information, and the third part (3. Inhibitory influences on sCP) that is in vague in some parts. We understand the author is trying to explain that research on this area (sCP) is lacking but it is not easy to read as compared to other parts of the manuscript (Rett part).

-       Part 3.2. is not well understood and readers could miss the point. References 226-233 are supposed to be related to CP but it is not clear. Please check it (lines 281-286). Anesthetics doses are not a main concern in guidelines or recommendations for surgery in patients with CP, so I would suggest reducing this part to a minimum. References are also not specific for this part (235, 236, 238).

Minor issues:

-       Please check the spell and/or meaning on lines 61, 143-145, 267-270, 346, 360, 370, 419, 426, 431, 474.

-       Figure 1 appears twice in the manuscript.

Author Response

REVIEWER 2

General Comment: This review approaches the interesting topic of inhibitory synaptic influences focusing on developmental motor disorders. The conceptualization and generous references lead to an attractive paper. However, it would be necessary to resolve some issues to improve its quality.

Response: We thank the reviewer for their encouraging response.

Major Comment 1: There is a gap between the second part (2. Inhibitory neurotransmission) that is full of detailed information, and the third part (3. Inhibitory influences on sCP) that is in vague in some parts. We understand the author is trying to explain that research on this area (sCP) is lacking but it is not easy to read as compared to other parts of the manuscript (Rett part).

Response: We agree and have changed the order of presentation to include Rett First – we have also added a section on the lanimal models and lack of adoption that hopefully brings more context to the sCP section.  

Major Comment 2: Part 3.2. is not well understood and readers could miss the point. References 226-233 are supposed to be related to CP but it is not clear. Please check it (lines 281-286). Anesthetics doses are not a main concern in guidelines or recommendations for surgery in patients with CP, so I would suggest reducing this part to a minimum. References are also not specific for this part (235, 236, 238).

Response: We recognize that this portion was weak, particularly with reference to the overall lack of clinical evidence and have removed this section

Minor Comment 1: Please check the spell and/or meaning on lines 61, 143-145, 267-270, 346, 360, 370, 419, 426, 431, 474.

Response: We have corrected these mistakes.

Minor Comment 2: Figure 1 appears twice in the manuscript.

Response: We have corrected the repeat.

Reviewer 3 Report

That managing developmental motor disorders can be equally challenging as learning disorders has been recognized only recently. It seems that coordinated therapeutic interventions might improve functional performance and ability to perform everyday tasks by those affected.

The knowledge on the inhibitory synaptic mechanisms at play in those disorders appears extremely important in this regard. It is well known that inhibitory deficits are present in cerebral palsy and Rett syndrome discussed in the paper.
Thus this manuscript may come out timely, dealing with so far intractable conditions to delineate research directions on which to embark.

The author does not seem to favor a particular inhibitory neurotransmitter e.g. GABA-ergic modulation for therapeutic intervention but instead postulates an integrative approach, stemming from diversity of inhibitory neurotransmitter targets including GABA and glycine interplay, as well as building on lessons learned from the analysis of the respiratory neuromotor system in this respect.

This is quite systematic review, with descriptions of neurotransmitter metabolism and receptor systems, ionic channels and transporters, pharmacological and clinical data as well as the corresponding developmental impact. The manuscript critically deals with the multitude of data and provides novel heuristic hypotheses thus adding novel insights to the literature.

However the author should elaborate some more on appropriate animal models of developmental motor disorders and/or the lack thereof. For example, whether animal models reflect the complexity of discussed motor disorders biology and to what extent; why there is no breakthrough in implementing novel treatments for those diseases based on the results of in vivo laboratory investigations.
The proposal of developing new animal models and what criteria they should fulfill would much strengthen the manuscript and would be met with great interest from researchers in this field.  

Other points: please, avoid common, nonscientific language throughout the text e.g. “The sheer abundance of transporter and receptor subtypes of the structures involved in GABAergic and glycinergic neurotransmission may be nauseating at first glance.”

Author Response

REVIEWER 3

General Comments: That managing developmental motor disorders can be equally challenging as learning disorders has been recognized only recently. It seems that coordinated therapeutic interventions might improve functional performance and ability to perform everyday tasks by those affected.

The knowledge on the inhibitory synaptic mechanisms at play in those disorders appears extremely important in this regard. It is well known that inhibitory deficits are present in cerebral palsy and Rett syndrome discussed in the paper.

Thus this manuscript may come out timely, dealing with so far intractable conditions to delineate research directions on which to embark.

The author does not seem to favor a particular inhibitory neurotransmitter e.g. GABA-ergic modulation for therapeutic intervention but instead postulates an integrative approach, stemming from diversity of inhibitory neurotransmitter targets including GABA and glycine interplay, as well as building on lessons learned from the analysis of the respiratory neuromotor system in this respect.

This is quite systematic review, with descriptions of neurotransmitter metabolism and receptor systems, ionic channels and transporters, pharmacological and clinical data as well as the corresponding developmental impact. The manuscript critically deals with the multitude of data and provides novel heuristic hypotheses thus adding novel insights to the literature.

Response: We thank the reviewer for recognizing the importance of the topic and displaying some enthusiasm for the approach.

Specific comment 1: However the author should elaborate some more on appropriate animal models of developmental motor disorders and/or the lack thereof. For example, whether animal models reflect the complexity of discussed motor disorders biology and to what extent; why there is no breakthrough in implementing novel treatments for those diseases based on the results of in vivo laboratory investigations.

The proposal of developing new animal models and what criteria they should fulfill would much strengthen the manuscript and would be met with great interest from researchers in this field. 

Response: We have added an addition section detailing the types of sCP animal models, their intent and their advantages and limitations. We add additional points in this new section regarding what criteria one would require form a good, or “least-worst” model.

Specific comment 2: Other points: please, avoid common, nonscientific language throughout the text e.g. “The sheer abundance of transporter and receptor subtypes of the structures involved in GABAergic and glycinergic neurotransmission may be nauseating at first glance.”

Response: We have toned down some of the nonsccientific language, particularly in the sections relating to scientific facts and knowlege.

Round 2

Reviewer 2 Report

Author have improved manuscritp significantly. Please check some minor spell mistakes along the text.